# The World Was Their Parish: Evangelistic Work of the Single Female Missionaries from the Methodist Episcopal Church, South, to Korea, 1887–1940

Angel Santiago-Vendrell [1,*] and Misoon (Esther) Im [2,*]

1 Asbury Theological Seminary, Wilmore, KY 40390, USA
2 College of Theology, Grand Canyon University, Phoenix, AZ 85017, USA
* Correspondence: a.santiago-vendrell@asburyseminary.edu (A.S.-V.); esther.im@my.gcu.edu (M.I.)

**Abstract:** The Woman's Foreign Missionary Society (WFMS) (1897–1909) and the Woman's Missionary Council (WMC) (1910–1940) of the Methodist Episcopal Church, South (MECS) worked in Korea from 1897 to 1940. Their work used a distinctive mission philosophy, hermeneutics, and implementation of strategies in their encounters with Korean women. Over the course of their years in Korea, Southern Methodist missionary women initiated the Great Korea Revival, established the first social evangelistic centers, educated the first indigenous female church historian, and ordained women for the first time in Korea. This article argues that, even though the missionary activities of the single female missionaries occurred in the context of "Christian civilization" as a mission theory, their holistic Wesleyan missiology departed from the colonial theory of mission as civilization. The first section of the article offers background information regarding the single female missionaries to help understand them. What motivated these females to venture in foreign lands with the Gospel? What was their preparation? The second section presents the religious, cultural, social, and political background of Korea during the time the missionaries arrived. The third section describes and analyzes the evangelistic and social ministries of the female missionaries in the nascent Korean mission. The final section describes and analyzes the appropriation and reinterpretation of the Bible and Christianity by Korean women, especially the work of Korean Bible women and Methodist female Christians in the quest for independence from Japanese control in the Independence Movement of 1919.

**Keywords:** world Christianity; Korean Bible women; Woman's Missionary Council; contextualization; social progress; missiology





## 1. Historical Background of the Missionaries Back Home

The central elements of the work of female missionaries of the Woman's Board of the MECS in Korea were based on the characteristics of Methodism and the unique theories of the Women's Missionary Movement in the U.S. The Methodist heritage that was adopted by the missionary women emphasized world evangelization, mobilization of itinerant preachers, class meetings, revivalism expressed in the camp meetings, Wesleyan theologies of social holiness, Christian perfection, and the holiness movement. The missionaries were also deeply influenced by the gender-based mission theories and strategies such as Woman's Work for Woman, Christian Home, and the spirituality of self-sacrifice that originated and was manifested in the Woman's Foreign Missionary Movement of the nineteenth century. In particular, the whole Methodist ethos that encouraged women's leadership roles in the Church worked in remarkable harmony with the gender-based missiology of the women's movement and contributed to the success of their labors in Korea.

This section examines some interrelated yet distinct historical backgrounds in North America which are most vital to understanding the female missionaries of the Woman's Board of the MECS and their work in Korea between 1897 and 1940, such as the organizational structure and mission history of the WFMS/WMC of the MECS, including that

of its denomination and its parent board, the General Board of Missions, the mission theories and strategies of the Woman's Board, the history of the Scarritt Bible and Training School/Scarritt College. This section also discusses the social, cultural, and religious background of Southern women. By examining North America in the context of Southern heritage and tradition, it is clear to what extent the missionary women of the MECS were restrained by conservative patriarchal norms and how significant therefore was the extent to which they improved their status. The missionaries themselves experienced an empowerment and transformation through their missionary beliefs and activities anchored in the Methodism and the WFMM of their homeland.

## 2. The Methodist Episcopal Church, South

The MECS was the second largest American Methodist denomination prior to the unification in 1939, when the Methodist Episcopal Church and the Methodist Protestant Church were united into the Methodist Church. The MECS originated in the Methodist Episcopal Church (MEC) and was organized as the world's first independent Methodist denomination at the Christmas Conference in Baltimore, Maryland, between 24 December 1784, and 3 January 1785. The first North American Methodist Church was a result of John Wesley's zeal to share the Gospel of Jesus Christ with immigrants to the New World. At the conference in Bristol, Great Britain, on 17 August 1771, Francis Asbury responded to John Wesley's call for volunteers to go to American; he became the first general superintendent or bishop of American Methodism. Thomas Coke, whom Wesley ordained and sent to America in 1784, was the joint superintendent and co-founded the denomination with Asbury (Barclay 1949). Over the decades, the MEC grew first in the South, then in the Midwest, New England, the West; it also expanded its missionary forces into Canada, Liberia, and South America. Then, in 1844, the Methodist Church split into two sections, with constitutional questions related to slavery as the primary dividing factor (Sledge 2005).

## 3. The Board of Missions (1845–1939)

When the denomination began, the Missionary Society of the MECS was organized to carry out missionary work both at home and abroad, and its first General Conference was held in 1846. In 1866, the missionary society of the Church was divided into a Board of Domestic Missions and Board of Foreign Missions; however, in 1870, the two branches were again amalgamated into one and governed by a general Board of Missions. In the General Conference of 1874, the Church amended the operational system of the missions by turning over the basic domestic missionary task to the Annual Conferences and all other missionary responsibility to the General Board (Clark 1925).

The administrative leadership of the early missionary organization and of the Board of Missions was held by the Secretary until 1921. In 1922, the office of General Secretary was ended, and the Board reorganized to include eight Administrative Secretaries. The Board restored the General Secretaryship in 1926; Willard G. Cram was elected to this office and served until the unification of the Methodist Churches. In 1939, the MECS ended its 95 years of history as an independent denomination by uniting with the MEC and the Methodist Protestant Church as the Methodist Church (Ferguson 1971).

## 4. Mission Theories and Strategies of the Woman's Board

The pivotal source of the mission theories and practices of the female missionaries of the Woman's Board primarily rested in their Methodist legacy. The missionary women shared some Methodist characteristic components with their contemporary male church leaders and mission personnel. In fact, the Southern Methodist missions possessed the mutual heritage of the MECS and the MEC even after their separation. Throughout its existence, the MECS carried out "a missionary movement that was distinctively Methodist, and not particularly Southern" (Sledge 2005). The Church adopted the traditional Methodist theology, motivation, tactics, language, and structure in their missionary ventures abroad,

and so the foreign missions of the MECS owed much to its Methodist tradition than its Southern context.

The mission philosophies and approaches of the Methodist women missionaries were surprisingly effective and successful in empowering and transforming both the indigenous women in Korea and other foreign fields for missions. Dana Robert pointed out, "some of the reasons for the overwhelming success of the Woman's Foreign Missionary Society . . . were the nature of American Methodism in the late nineteenth century" (Robert 2000). According to her, peculiar factors, which contributed to the flourishing missions of this women's board rooted in Methodism, included optimistic Arminian theology, strong laity, its system of circuit riders, class meetings and societies, empowerment of women, and theological emphasis on human cooperation in one's own salvation and personal religious experience (Robert 2000).

The gender-based mission ideologies and methods of the Protestant Woman's Foreign Missionary Movement, including Woman's Work for Woman and the Christian Home, both demonstrated and enhanced the effectiveness of the missionary purpose and the strategies of the Woman's Board of the Southern Methodist denomination. Examining the mission theories and tactics of the Woman's Board of the MECS in North America is paramount for understanding the historical background of the missionary women in Korean society because these women missionaries applied the mission philosophies and principals to their endeavors in that mission field. By employing the knowledge and experiences they had gained in their homeland, the Methodist female missionaries in Korea carried out their missionary efforts with aplomb and confidence.

A valid criticism against the Methodist Episcopal Church, South in general, as well as the single female missionaries, was their position on slavery; their pro-slavery viewpoint had been the original division point for the Church. The position of the MECS regarding slavery was clear: slavery was divinely approved by God. William Gannaway Brownlow's sermon to the church's assembly of 1852 sanctioned slavery as divinely inspired. He claimed, "Civil rights are never abolished by any communication from God's Spirit; and those fiery bigots of the North who proposed to abolish the institution of slavery in this country, are not following the dictates of God's Spirit" (Brownlow 1857). The female missionaries were women of their time, and it took time for them to denounce slavery. It was not until 1897 that Bennett began mentioning the demand for organized missionary work among black people; she started to continually encourage the Methodist women to become involved with interracial settlement work. She stated, "There are two large classes of society touching our lives every day, whether we be in the country, in the towns, or in the large cities, to which our Society has yet given no helping hand. We have done nothing for the negroes. Our great Church, as a Church, has done but little. Individual members may, doubtless, have done much; but as an organization the path of duty is plain before us . . . . We must enterprise some special work for this great race of people" (Bennett 1897). In 1912, the WMC members at last responded to Bennett's call to action by launching settlement work for black people, and in 1913, founded the Nashville Bethlehem House in Nashville, TN. The Woman's Board also established additional Bethlehem Houses in Augusta, Georgia, Birmingham, Alabama, and Chattanooga, TN. The Southern Methodist women in the North America soon began to extend their contribution more to the public realm by employing an all-out social gospel program that expressed a strong social concern for the marginalized of the society. In 1916, the project discussed "the abolition of child labor, the reduction of illiteracy, prison reform, and end to the convict lease system, the cultivation of 'sympathy between all races,' and the solution of race problems in a spirit of helpfulness and justice" (Scott 1970).

## 5. The Scarritt Bible and Training School/Scarritt College

By the late nineteenth century, the leaders of the Southern Methodist women's missionary organizations had been keenly aware of the need for an educational institute to train female ministry workers both at home and abroad. In 1887, Miss Belle Bennett began

to consider establishing a training school for missionary women. In 1892, the Scarritt Bible and Training School for Missionaries and Other Christian Workers opened in Kansas City, Missouri.

The purpose of the Scarritt School was "to train not only foreign missionaries, but also all Christian women who seek to fit themselves for the service of God anywhere, including nurses, for which purpose a small hospital is one department of the institution" (Gibson and Haskin 1928). The educational program and curriculum of the school was well integrated to prepare the women to effectively carry out various educational, social, and medical labors along with evangelistic ones (Gibson 1901).

However, the school essentially functioned as if it had only two departments: the Bible Department and the Nurse-Training Department. Each department required two years of study for graduation. The Bible Department emphasized the importance of studying the English Bible as the basic text. Students in the Nurse-Training Department were required to receive basic training of elementary medicine and nursing, study health laws, and learn methods of domestic work; these were considered required fields of study to prepare students to take care of children and teach people in the mission fields. The student had to participate in city mission work that aimed to prepare her to go mission fields, where she would confront worse conditions. Through such discipline, the training school pushed for the missionary candidate to become realistic and dismiss her "visionary or romantic views of mission life," diminishing "self-conceit" and "self-seeking" while learning to live with "self-knowledge," "self-reverence," and "self-control" (Cobb 1987).

## 6. Southern Women

Before they had learned to stand up for their own right to preach the salvation of Jesus Christ and implement social holiness among women around the world, the missionary women, and their leaders of the WFMS in the MECS had, for generations, been socially and culturally bound by the pervasive conservative patriarchal structure of their society. This section provides a preliminary understanding of how the experiences of the Southern Methodist missionary women in their homeland could engender in them such empathy for Korean women; Korean women's situation was arguably much worse in their conservative patriarchal Confucian society. It is certainly understandable that the female missionaries, having themselves experienced such transformation through their belief in Christianity and through education, would want to share not only their faith in Christ but also Christian education and other social outreach activities with Korean women. It becomes clear that Christianity, and particularly the whole Methodist ethos, empowered women both in the U.S. and foreign mission fields.

The Southern social structure was hierarchical because it was an agricultural and slave society, and every southerner, white or black, male or female, rich or poor, had their assigned place. Men, women, and slaves were intrinsic parts of the patriarchal system, and they were expected to support and sustain the patriarch, typically by working to create the ideal plantation, which was controlled by the patriarch (Scott 1991).

The prevalent Southern evangelical theology reinforced this hierarchical society. By the early nineteenth century, white Evangelicals in the South, including Methodists, believed in the absolute authority of men over their dependents. Evangelical writings about the relationship between domesticity and female piety and the question of a woman's sphere swept through Anglo-American evangelical circles and reinforced such beliefs. In short, by 1830, women were viewed as emotional and gentle, inferior to men, and incapable of receiving—or indeed having need for—higher education; thus, when a woman attempted to pursue a profession, she was deemed unworthy and masculine. As Scott states, "Nowhere was this more evident than in the American South" (Scott 1970).

Upper-class Southern white women were more strictly limited and deeply entrenched in this image of a pious woman and the concept of female domesticity than Northern women. From very early childhood, girls were disciplined to be submissive and to obey men, as described by evangelical biblical interpretation. Methodists and Baptists of the

South especially demanded perfection in Christian women, and these women internalized this demand in their family lives. The intense socialization of such beliefs meant that it took Southern women much more time and effort to free themselves from this oppressive social restraint and to find a path to self-determination.

By the 1850s, however, Southern women became aware of the news about the woman's right dispute (a movement with beginnings in the North during the 1830s), and through the Civil War, Southern women were not seen as a threat to the Southern patriarchy, but rather as an essential part of that society. They worked to sustain their society in the absence of their menfolk by running the plantations and supervising the slaves. In so doing, of course they began to realize their abilities were far greater than they, or society (meaning men), had allowed. As a result, women assumed more public responsibilities and gradually changed their relationship to the church. Of course, these changes were hardly dramatic or egalitarian; Southern women were still more likely "to work in 'woman's sphere' quietly cooperating with the men in social and economic rehabilitation, but wherever necessity required they publicly displayed a varied executive ability" (Mendenhall 1934).

### 7. The Religious, Cultural, Social, and Political Background of Korea

When the missionaries arrived at Korea in 1887, several religions existed, including Shamanism, Taoism, Buddhism, and Confucianism. Considering itself to be a "small China," Korea adhered to the Confucian tradition more strongly and for longer than China and remained the last Asian country to open its doors to the West in 1882.

By and large, Korean religious history can be divided into five periods: the Shamanistic period (pre-fourth century), the Shamanistic-Buddhist period (372–1392), the Shamanistic-Buddhist-Confucian period (1392–1784), the Shamanistic-Buddhist-Confucian-Roman Catholic period (1784–1884), and the Shamanistic-Buddhist-Confucian-Roman Catholic-Protestant period (1884–present). Even though Taoism and Tonghak have never dominated a period, the former has existed from the second period on, and the latter from a time between the fourth and the fifth periods (Oak 2006).

Shamanism is the oldest religious tradition in Korea. Even though some scholars, such as Eliade and Clark, assume that Korea Shamanism originated in Siberia and Inner Asia, it is difficult to determine the actual origin of the religion. Shamanistic religious life centers on the shaman or mudang, who plays a role as a mediator between humans and the gods, spirits, and nature. Throughout thousands of years of their history, Koreans blended Shamanism with other forms of magic and other world religions, such as Buddhism, Taoism, and Confucianism. Shamanism was also influenced by folk music, dance, drama, and many folk customs which combined to become the culture of ordinary Korean people (Hulbert 1962).

Taoism was introduced to Korea in the fourth century. Without proclaiming itself as a religion per se, Taoism was blended with Shamanism, Buddhism, and Confucianism, and over many centuries, it was subsumed as a folk religion in the many forms of religious practices that existed in Korea. Korean Taoists believed in Shangti and his trinity, and that human beings could be mystically united with God through sincere prayers. Adherents particularly emphasized "ethical life, physical strength and health, and long life; and their holistic approach aimed at a sound mind in the sound body" (Clark 1925). The Yi dynasty, however, persecuted Taoists as heretics; only those who were societal outcasts embraced the religion. In later years, Protestant missionaries in Korea did not pay as much attention to Taoism as they did to Shamanism and Confucianism (Yamashita 1996).

The historic religion of Buddhism, which originated in India, came to Korea by way of China, and provided the Korean people with a transcendental spiritual doctrine. Buddhism provided the spiritual inspiration for the aristocratic artistic works of the Three Kingdoms period of the Silla (f. 37 B.C.), Koguryŏ (f. 37 B.C.), and Baekchae (f. 18 B.C.) dynasties. This period ended in the seventh century, and being a state religion, the Contemplative School or *Sŏn* Buddhism flourished during the Koryŏ dynasty (918–1392). Buddhism soon lost its strong emphasis on transcendental spiritual aspects and focused on more secular

matters, absorbing beliefs and practices of Shamanism and other folk religions. Therefore, by the end of the Koryŏ dynasty, Buddhism was no longer meeting the spiritual needs of the populace and society (Hulbert 1962).

In the Yi dynasty that succeeded the Koryŏ, Confucianism from Mongol China became dominant. It did so by lending its orthodox religious ideology not only to the state (for example, in its governing principles) but also to the people (for example, in the family structures and ancestor worship it advocated). Though a reforming force and the state sponsored religion of the Yi dynasty, Confucianism also lost its original vitality, spiritual meaning, and mass appeal, and became merely a formal tradition (Chung 1997).

The first Roman Catholic priest who ever entered the Yi dynasty and stayed for two years was Gregorio de Cespédès. However, the actual origin of the Korean Catholic Church began some decades later, when a few young and famous scholars, such as Chŏng Yak-chŏn and Kwŏn Ch'yŏl-sin, first read some Chinese language Christian tracts, including Matteo Ricci's *True Doctrine of the Lord of Heaven*, and then began to practice what they learned in the booklets. In 1631, the Korean ambassador Chŏng To-wŏn brought the *True Doctrine of the Lord of Heaven* to Korea for the first time. In 1783, the scholar Yi Sŭng-hoon went to China and was converted and baptized as Peter Yi after meeting a priest there. In 1784, he returned to the dynasty, began to convert people, and baptized them. New believers steadily increased in number throughout the country; by 1863, there were more than 19,000. By 1884, there were more than two thousand martyrs in the country; and by the end of the century, Catholicism in Korean was forced underground (Hulbert 1962). This long and varied religious tradition faced the late nineteenth-century pioneer Protestant missionaries, including the American Methodists, when they came to Korea.

## 8. Socio-Political Background

By the late nineteenth century, when American Protestant missionaries entered the Chosŏn or Yi dynasty (1392–1864), the dynasty had been known as the "Hermit Nation" because of its policy of seclusion. The government was a despotic monarchy and, by that time, was under the regency of the Taewŏngun, who adhered to the foreign policy of adamantly excluding and violently opposing foreigners and all things foreign; this opposition was particularly focused on the Christian Church. The Korean king and people believed and practiced the Confucian philosophy that had been the state religion for nearly 500 years and regarded Christianity as an evil cult (Hulbert 1962).

Meanwhile, the socio-political situation of the period worked in favor of Christianity entering Korea as a new religion. By the time the Protestant missions launched their work in Korea, high imperialism prevailed over the politics of the Far East nations. The political struggles among China, Japan, and Tsarist Russia played a crucial role in bringing Christianity into Korea. These nations bordering Korea were the three major powers in north-eastern Asia. Since the Meiji Restoration in 1868, Japan had implemented a policy of expanding her territory toward the Chinese continent and had developed strategies to use Korea as the stepping-stone to fulfill their ambition (Stokes 1934).

In February 1876, Japan and Korea signed the Treaty of Kanghwa that was unequally imposed by the former country and ostensibly articulated her acknowledgement of Korea's independence from other countries. Japan imposed this unequal treaty on Korea, just as the United States had on Japan in 1853, and European countries had on China in 1842. Through this treaty, Japan had the right to compel the Korean court to grant any prerogative to Japan without any intervention from China. Leaders of the Chinese government, such as Li Hongzhang, advised the Korean court to negotiate other mutual treaties with Western countries to attenuate Japan's influence in Korea. The Shufeldt Treaty was concluded between Korea and the United States in 1882, and similar treaties immediately followed between Korea and England, Germany, Italy, Russia, and France. The treaty between Korea and the western countries allowed foreigners to trade and build residences at the open ports of Korea. As a result, China's claim of suzerainty over Korea weakened, and Japan's power in the country was enhanced (Moose 1911).

As Korea entered treaties with foreign nations, she attempted to embrace the idea of modernization while retaining her independence; however, her sociopolitical systems were violently shaken up, not only because of the effect of her longtime isolationism in foreign policy but also because of continued interference by foreign powers. Throughout the subsequent years of power struggles among China, Japan, and Russia, Japan finally won victories over China in the Sino-Japanese War in 1894 and over Russia in the Russo-Japanese War in 1904. Japan therefore held exclusive power over Korea, gaining the right to use Korea as a path to Manchuria, and guaranteeing Korea's independence and the safety of the imperial family as concessions. The Protectorate Treaty between the two countries was signed in 1905, King Kojong abdicated in 1907, and the Japanese annexation of Korea began in 1910. Japanese control in Korea continued until 1945, when Korea gained her independence (Gale 1898).

### 9. Arrival of the Single Missionary Women

Between 1897 and 1938, the WFMS/WMC sent 74 single women missionaries to Korea (*ARWFMSMECS*). The "Requirements of Missionary Candidates" clarified that the candidate must be less than 25 or more than 35 years old. Even though it did not mention that the missionary candidates "must" be single, "Instructions to Missionaries" emphasized that "missionaries [female workers for women's work] must give [their] entire time and attention to their legitimate work" (*ARWFMSMECS*). Thus, missionaries who married had to leave their positions as official missionaries of the woman's organization.

On 9 October 1897, Josephine Peel Campbell, the pioneer missionary of the WFMS of the MECS, arrived in Seoul to begin the denomination's full-scale women's work. When she entered Korea, Campbell was accompanied by her adopted Chinese daughter Dora Yu. Yu was a medical doctor; she had graduated from Soochow Medical College in 1896 and served in the Mary Black Memorial Hospital established by the WFMS of the MECS. When Campbell and Yu arrived in Seoul, Rev. Reid and his family, who had also been in China and knew Campbell, gladly welcomed them into their home (Campbell 1897–1920).

Influenced by the Methodist legacy that emphasized one's evangelistic responsibility and the philosophy of Woman's Work for Woman, Campbell's ultimate concern in her missionary labor was to save the souls of Korean women and children. To reach this goal, she first wanted to focus on establishing an educational system for women and girls that she felt would be a great means to evangelize Korean people. She stated,

> Our plan for the work under the Woman's Board is, as soon as a speaking vocabulary will permit, to open a class for women and girls, which we hope may be done during the coming year . . . . The Bible will be the principal textbook . . . . We are longing to see the Church on its knees for the conversion of Korea. If the Christian world could only, for themselves, see this heathen world as Christ sees it our hearts would be quickened to duty, and it would not be what shall we eat, what shall we drink, and wherewithal shall we be clothed, but how can we save a soul for eternity. (Campbell 1897–1920)

In addition to working for the development of mission programs and institutions in Seoul, Campbell also launched missions in other major cities in Korea, such as Songdo, Wŏnsan, Ch'unch'ŏn, and Chŏlwŏn. Along with Seoul, Songdo (or Kaesŏng) of Kyŏnggi Province was another stronghold of the Southern Methodist mission in Korea under the leadership of Arrena Carroll and Fannie Hinds (Campbell 1897–1920).

The institutions which the Southern Methodist missionary women in Korea established and ran for over forty years included four boarding schools for girls and women, three Bible schools for women, several women's night schools, and numerous day schools and kindergartens for children in their mission stations of Seoul, Songdo (Kaesŏng), Wŏnsan, Ch'unch'ŏn, and Chŏlwŏn. The Southern Methodist missionaries also cooperated with Northern Methodist missionaries by getting involved in the teaching and administration of well-known educational institutions, such as Ewha Woman's College and Kindergarten

Training School, and the Methodist Woman's Bible Training School that developed into the Union Methodist Theological Seminary.

The institutional mission history recorded by Southern Methodist female missionaries reveals that the missionaries emphasized three aspects of their mission beliefs. First, the missionaries always regarded evangelization of souls as the most significant goal of their missionary social outreach. Second, even so, the missionary women contextualized the message of Christ by employing holistic mission approaches that highlighted their educational, social, and medical services. Finally, believing a Christian home to be the cornerstone of Christian society, the missionaries put great efforts into establishing a Christian home for each family through their institutional missions for Korean women, girls, and children.

These dual concerns of evangelization and social outreach through holistic mission approaches and the central role of the Christian home arose out of the emphases of Methodism and the WFMM in their homeland. Historically, following the example of John Wesley, Methodism had always emphasized the evangelistic responsibility of the Church. The Wesleyan tradition understood Christian social services as a fundamental element of evangelism, displayed through its theologies of social holiness and Christian perfection that stressed not only evangelistic demands and individual holiness, but also social responsibility.

Another foundation of the Southern Methodist female missionaries' work in Korea was the mission theologies of the late-nineteenth century WFMM. In addition to the optimistic Methodist theologies, these gender-based theories provided a strong impetus to the missionary women to implement their holistic approach in the mission field. Woman's Work for Woman and the Christian Home formed the two pillars of their theories. These emphases, according to Dana Robert, meant that American women missionaries historically carried out holistic work, "balancing evangelism with the meeting of human needs through education, health care, and social transformation, and avoiding the dichotomy of 'Christianizing' versus 'civilizing' that runs through the history of mission thought as usually depicted" (Robert 1997).

This goal was quite intentional, as the primary educational purpose of Scarritt Bible and Training School/Scarritt College from which the missionaries had graduated was to train Southern Methodist women to work among women and children around the world. It was at Scarritt that the missionaries learned how to perform their holistic missionary labors, including various educational, social, and medical activities as well as evangelistic work. This synthesis of the two historical movements of Methodism and the WFMM empowered the Southern Methodism women missionaries to achieve success in their mission programs and institutions in the Korea mission field.

The women missionaries primarily educated women and children within the four major forms of educational institutions: boarding schools for girls and women, Bible institutions for women, day schools (*kŭlbang* or *sŏdang*) for children, and kindergartens for small children. At the time, it was improper for middle or upper-class girls to walk on the streets, which was one reason that the Woman's Board primarily founded boarding schools for girls.

Even though the students at the earliest boarding schools established by the missionaries also came from poor families, missionary schools depended on tuition for their operating costs. However, missionaries did provide scholarships and offered so-called industrial programs through which girls could earn money for their school expenses. Over the years, the boarding schools expanded to include primary and high schools. From the beginning, Southern Methodist women missionaries in Korea, such as Campbell, also developed Bible schools or theological schools for women to train indigenous workers such as Bible women for the mission field.

Southern Methodist mission schools for girls and women usually provided industrial training to their students. Such training typically included foreign sewing, knitting, crocheting, embroidery, and foreign cooking classes. This training was important for several reasons. First, the missionaries wanted female students to be able to make a living after

graduation as well as contribute to their self-support during the school year. Second, the missionaries believed that such industrial training could help the girls and women to be better homemakers. Third, this training helped the missionary schools compete with Japanese government schools that provided various kinds of industrial trainings to their students. Mission schools sometimes employed Chinese or Japanese teachers to teach specialized industrial courses. During this period of Japanese control, many Korean people worked as servants for Japanese people and companies. With the skills and licenses gained at school, after graduation, Korean girls could work for the Japanese as industrial workers and teach industrial subjects in any private school.[1]

Along with boarding schools, day schools, including one-room schools, played a significant role in the growth of the church in Korea during the pioneering years of the Southern Methodist Mission. When the Protestant missions began, the churches took over this existing school system, modifying its curriculum by adding arithmetic, geography, hygiene, and religious teachings, and notably establishing this type of schools for girls as well as boys. Day schools flourished in rural areas in particular because of the schools' close relationships with the country churches. Such schools provided education for the children of Christian parents who could not send their children to boarding schools (Wasson 1948).

As the Japanese government demanded that the mission schools be run according to its regulations, the Woman's Board in the United States increasingly wrestled with the financial consequences of providing for separate educational and religious work. Owing to the lack of governmental restriction upon kindergartens, the women's missionary society eventually focused its efforts on educational work in kindergartens, which rendered a vital service especially in gaining access to non-Christian homes (Tatum 1960).

The Southern Methodist women missionaries used their experience in their homeland—particularly of Methodist churches or groups—as a model for their educational work with Korean women and girls. In the schools that they founded, the missionaries organized such religious activity groups as Bible classes, the Sunday School Board, Epworth League Board, and the Young Woman's Christian Association (YWCA), along with missionary societies and debating societies (Rhie 1990). Through participation in these religious activities in the schools, the students developed their own spiritual discipline and leadership skills and played significant roles in evangelistic and educational work among other Korean women, girls, and children.

One concern of the women missionaries for their schools that was deeply rooted in Methodist tradition was to have ready access to the indigenous people for the purposes not only of teaching them various academic disciplines but also of evangelizing them through educating girls and women. Influenced by the Woman's Work for Woman, the missionaries also hoped to improve the status of these women. The missionaries believed that education could help women enlighten and liberate themselves from the miserable conditions under which they lived, conditions determined by a very conservative Confucian philosophy, social, and familial system. Educated in the mission schools that persistently emphasized the importance of the Christian Home through their curricula and programs, many of the graduates of the mission schools got married and became Christian homemakers. Others became teachers in country or rural schools, Bible women in local churches, social workers, and nurses in mission hospitals, all of which contributed enormously to the spread of Christian influence in their society. The Woman's Board of the MECS established and ran such schools for women and girls in strategic places—mainly in cities such as Seoul, Songdo, Wǒnsan, Ch'unch'ǒn, and Chǒlwǒn, where its mission stations were located.

The Southern Methodist female missionaries became the pioneering social workers among women and children in Korea. Their social programs and labor in both rural areas and cities were creative and unique among Protestant missions. In the countryside, the missionary women formed clubs, classes, and unions to teach and spread helpful agricultural methods and technologies. In cities, they provided various social services in their social evangelistic centers that were a unique feature of the Woman's Board of the

Southern Methodist Mission. Some of the women missionaries also played leadership roles in the non-denominational WCTU in the mission field.

In the early years of their work in Korea, the Southern Methodist women missionaries did not perform specific social work tasks. A large portion of the women were in so-called evangelistic circles, and they regarded rural evangelism as very important. The Woman's Board appointed them as district workers. These district evangelical missionaries usually carried out some individual or co-operative projects which enriched life for the people in addition to their own evangelistic ministry. The missionaries served the indigenous people in every possible way, utilizing their talents and skills for the benefit of Korean society. Often a public health nurse, a doctor, and a preacher made a team to serve as many people as possible. By sacrificing their own private lives and spaces, the women missionaries also provided their own homes as public meeting places for the people of the towns or villages. For example, Laura Edwards and her Bible women turned their house in a small village outside Seoul into a "neighborhood house." Some of the neighbors came to the house to attend clubs or classes in the afternoons and evenings while others worked with different groups in local churches (Tatum 1960).

The program of the Union included such subjects as raising chickens and pigs, bee-keeping, cooperation in buying and selling, soil-testing, seed selection, and use of fertilizers. Most of the participants were Christians; however, others were also welcomed, enjoyed the activities offered, and came to have a much friendlier attitude towards Christianity as a result. Some of these non-Christian attendants at the Sunday services, inspired by their experience in the institute, decided to accept Christ (Tatum 1960). Thus, through their rural social work, the Southern Methodist female missionaries not only helped the Koreans to improve their economic situation and living circumstances but also tried to win individuals for Christ. Because of their concern for evangelization of souls, the missionaries never thought of their social work as being separate from their evangelistic endeavors.

Southern Methodist women missionaries in Korea also pioneered urban social work by establishing social evangelistic centers that closely resembled the settlement houses in their homeland. The WMC provided Centenary funds to begin the project of establishing the centers in the mission stations. While focusing on the twofold (social and evangelistic) missionary work in the centers, the missionaries carried out educational and medical work as well. These multi-purpose centers offered education, a place to socialize through various societies, clubs, and other activities, and medical work for women and children in their clinics.

## 10. Korean Bible Women: The Premier Evangelists

Throughout the annual reports of the Southern Methodist missionary women to the Woman's Board, comments about the labors of Korean Bible women who played a crucial role in evangelizing their fellow countrywomen abound. In the pioneering years, two Bible women, who were trained by the women missionaries of the Northern Methodist Mission, provided great support to both Campbell and other women missionaries.

Bible women were probably among the Korean women's groups that had the most influence of the missionary women. They were always educated, trained, appointed, supported, and supervised by the missionaries while performing their duties as missionary assistants. From the beginning of the Mission in Korea, the Woman's Board of the Southern Methodist missionary women acknowledged the significance of educating Korean women workers to produce Bible women, and the pioneer missionaries such as Campbell, Yu, and others in Seoul, Songdo (Kaesŏng), and Wŏnsan opened Bible classes for women to meet this goal.

Bible women in Korea consisted of both paid and volunteer workers, with paid workers supported by scholarships from the Woman's Board of the Southern Methodist Church. The missionary board usually provided scholarships to students who were preparing to be Bible women in the Southern Methodist mission schools, such as Alice Cobb Bible School, Joy Hardie Bible School, Holston Institute, and Mary Helm School (Carroll 1897–1919).

Southern Methodist women missionaries whose primary responsibility was evangelistic work always had Bible women as their assistants, and it was these Korean Christian workers who made actual contact possible between the missionaries and their non-Christian sisters in villages and countries. There were always an insufficient number of missionaries in the districts in comparison to the tremendous work they took on. The missionaries trained and supervised Bible women for this reason.

Under the supervision of the female missionaries, Bible women focused exclusively on evangelizing their female compatriots not only in local churches but also in other Southern Methodist institutions, such as schools, hospitals, and social evangelistic centers. Beyond this, Bible women carried out house-to-house visitation, itineration trips, and taught Christian women in local churches. When they traveled with women missionaries, the indigenous women played the roles of translators and mediators. The Korean Christian workers also significantly contributed to the promotion and establishment of Christian homes through their visitation and itineration under the instruction of their supervisor missionaries (Carroll 1897–1919). Bertha Tucker reported, "The Bible woman is expected to visit the Christian homes in her circuit as often as possible, to help the Christian women with the Bible Course of Study and teach those who cannot read" (Tucker 1988). In the churches, the female Christian leaders taught some women to read and others in Bible or catechism classes. Through their evangelistic work, Bible women were not only evangelists but also agents of education and literacy.

The evangelistic labors of Bible women in the schools included visiting and preaching in the homes of boys and girls of the day schools and supervising those schools. In the Southern Methodist mission hospitals, the Korean Christian female leaders focused on preaching and evangelizing the patients and their families (Carroll 1897–1919).

The Bible women's role was also crucial in the areas of social work of the Southern Methodist women's work. First, the Christian leaders worked among village women introducing recreation programs; teaching women to read and write the Korean alphabet; helping them to manage the family finances; and instructing them how to raise silkworms, pigs, chickens, rabbits, and cows; and other "practical subjects such as hygiene, sanitation, first-aid, baby care, budgeting, dyeing, sewing, [and] cooking" (Induk 1935–1936).

Second, Bible women's work in the social evangelistic centers was essential, as Tatum reminds us: "No record of the social-evangelistic program in Korea is complete without mentioning the native Bible women who were indispensable in the work of missions, both in the cities and in rural areas. Their knowledge of the language and customs of their people made them natural leaders and opened doors of opportunity rarely available to the missionaries. Their understanding of the Oriental mind and their own experience of conversion placed them in a unique position to win followers for Christ" (Tatum 1960).

In the 1920s and onwards, when Korean women's missionary societies had begun to flourish, the workload of the female Christian leaders increased as they took on duties in the women's organizations, including leadership roles in Bible study classes, morning watch societies, and the Methodist Laymen's Movement (Tucker 1988).

Such tasks and responsibilities, one might say, are not significantly different from those of women pastors in modern Christianity. Certainly, they were much appreciated, as Sadie Moore's report on her Bible women and their tasks in Wŏnsan District attests:

> How I should like to be able to take you with me on an itinerating trip over our Wonsan District and let you have a peep in on the work that our twenty-three faithful Bible women are doing to keep up the spiritual life of our Christian groups.... Each woman has several churches on her circuit, and she goes from one to another, visiting in the homes of the people, holding services, inviting people to church and exhorting people everywhere to believe on the Lord Jesus Christ. The Bible Women are the inspiration and leaders of the missionary societies, help in the work of the Sunday-school, very often take charge of the preaching service on Sunday, they visit and minister to the sick, they help plan and arrange for

funerals and weddings, in fact they stand ready to help in every activity of the church and community. (Moore 1960)

Moore's appreciation was not an exception. The annual reports of the Southern Methodist missionary women to the Woman's Board contain numerous appreciations and compliments for the indigenous Bible women and their labors. The comments of the missionaries on these Korean female Christian workers affirmed the essential roles of the latter in missionary work of the Southern Methodist women among indigenous people. For example, in 1919, Bertha Tucker recorded, "We could not travel without her [a Bible woman traveling with her]" throughout 26 preaching places in the West Wŏnsan District of the Southern Methodist Mission (Tucker 1988).

There were many other historical events that demonstrated the significance of Bible women's work. When the women missionaries left to take a leave of absence, Bible women performed the work of missionaries. In the annual report of 1902–1903, Fannie Hinds of Songdo reported that, while she was out of town, her Bible woman was "a splendid substitute for a missionary, kept the work going, so nothing was lost to the work," and she gained much. Sadie Moore's writing spoke of the outstanding spiritual or inner qualities of her Bible women as well as their contributions in evangelizing people:

> One of the great joys and privileges of my life to get to know and work with these dear women, who are indeed soldiers of the cross. Many of them work under very hard circumstances and yet I am sure you could not find a more a beautiful spirit of service and devotion to the cause of Christ than is manifest in the life and work of these women. A missionary friend remarked the other day that the Bible Women will surely have a high seat in heaven. Well, I am sure [they will] hereafter, but I am so glad that they are also reaping rewards every day. God is blessing their efforts in a very wonderful way, new believers are constantly being led to Christ, and believers are being built up in the faith through the faithful ministry of these women. (Moore 1960)

Thus, the Korean Methodist Bible women learned the Christian faith and skills needed for essential holistic evangelistic work for women in various contexts from the Southern Methodist female missionary mentors. Under the guidance and influence of the missionaries, the indigenous Christian workers, who became the invaluable assistants of the women missionaries, greatly contributed to evangelizing their fellow women, and spreading the ideas of Christian homes and social services. The Southern Methodist female missionaries originally adapted this concept of Bible women from the deaconess movement in their homeland. Women's foreign missionary societies in the United States employed the notion of the deaconess in various mission fields around the world, and among the Protestant women's missionary groups, the Methodists deftly applied the concept in their own mission fields. In Korea, the Methodist women successfully employed and developed the system of Bible women (Montgomery 1920).

Deeply influenced by the missionaries of the Woman's Board of the MECS, Korean women made decisions mirroring the female missionaries in several ways. Korean women were also determined to devote themselves to Jesus Christ and the Kingdom of God, following the examples of the lives of the individual female missionaries; thus, they created new professions and roles in their society as teachers, social workers, and other civil leaders, as well as fulltime church workers.

Some of those first pupils went on to creatively apply Christian doctrines of liberation not only to their own lives but also to their community, nation, and foreign lands. Korean women were vigorously involved in the 1919 Independence Movement and initiated foreign missions just a few decades after the missionaries of the Woman's Board of the MECS entered the country. These historical events attested to how Korean women were transformed because of the gospel message and the modern education brought by women missionaries, as well as to the indigenous women's ability to implement and contextualize

the message. A good example is the influence of Christian women in the Independence Movement of 1919.

## 11. Independence Movement of 1919

In the years between 1910 and 1945, when Korea came under Japanese control, although U.S. missionaries followed their mission board's policy and avoided confrontation with the government, Korean women were noncompliant and protested the political authorities despite strict punishments that included imprisonment, torture, and even the death penalty. When the Independence Movement emerged in 1919, indigenous Methodist women leaders and the students at the Southern Methodist mission schools including Paewha in Seoul and Holston in Songdo were heavily involved. Because of the deep engagement of these students in that event, the educational work of the Southern Methodist women missionaries was severely interrupted.

Paewha students, especially those who were the members of YMCA/YWCA in Seoul, actively expressed their patriotism. As a preliminary action of the March First Movement, they secretly brought hundreds of copies of the Declaration of Korean Independence that were copied in the basement of Ewha Haktang to their school. Ewha, which was established and run by the women missionaries of the Northern Methodist Missions during that time, was the cradle of the earliest women's national movement in Korea. The story of one of the students, Ryu Kwan-Soon, a Korean Jeanne d'Arc, is well known (Rhie 1991).

In the days before 1 March, the Paewha students secretly stored, publicized, and posted copies of the Declaration in the stores and houses around the area of the school during the night. Even though the students were gravely disappointed to learn on 3 March that the school authorities were prohibiting them from joining the movement, they had an independent position that differed from the missionaries.

At daybreak on the day of the first anniversary of the March First Movement, the 40 Paewha students went up to the highest point of the school (where they could look down on the city) and praised Korea by waving its national flag that they had secretly prepared over the previous days. Japanese authorities immediately arrested 25 of the students, and pressed the president of the school, Miss Smith, to take responsibility for the girls' actions. Smith responded that the reason she had come to Korea was not political but educational, and that she did not want to punish students who were encouraging the independence of their own country. As a result, one student was imprisoned, and the other 24 were released on condition that Miss Smith resign her presidency of Paewha (Chang 1934). Chang Chŏng-sim, the first Korean female Church historian to publish the first Methodist Church history, was another prominent Southern Methodist Korean woman who contributed to promoting a patriotic sprit. She was likewise converted by Southern Methodist missionary women, and graduated from Southern Methodist mission schools including Holston, Ewha, and the Methodist Woman's Union Bible School. After experiencing the March First Movement, she became an influential Christian national poet under the Japanese regime while carrying out educational work for women in Songdo and Seoul (Rhie 1991).

## 12. Conclusions

Dana Robert asserted that world Christianity has been a women's movement in its history, and thus "studies of world Christianity, either as global force or as local movement, need to put women's issues at the center of our scholarship about the growth of Christianity in Asia, Africa, and Latin America" (Robert 2006). In this paper, we argued that the female missionaries of the WFMS/WMC of the MECS between 1897 and 1940 in the Korea mission field played a significant role in shaping Christianity as a women's movement.

The holistic mission belief of Southern Methodist missionary women anchored in the two historical legacies of Methodism and the WFMM enabled the missionaries to adopt some elements of progressivism into their educational mission method. For example, influenced by progressive educational philosophy of the late nineteenth and early twentieth centuries, the missionaries employed scientific educational methods that emphasized the

development of human potential and personality and character-building by carrying out holistic education that focused on physical, mental, intellectual, and spiritual development. This educational philosophy also provided the concept of human equality.

The holistic missionary women also adopted some elements of progressivism into their belief in the Christian Home that was traditional, and thus contributed to the modernization of the country. In the early 1910s, the missionary women adapted scientific homemaking or home economics into their missiology and strategy of Christian Home, and taught scientific methods of infant care, nutrition, hygiene, and homemaking in Korea. Since the family structure of the Christian Home emphasized the value of an educated woman, women emerged at the center of the family owing to the changing definition of an ideal woman in the family and, by extension, in society.

Such education provided by the missionaries empowered their indigenous students to see themselves as the subjects and prime movers in history. Influenced by the teachings of the missionaries, the self-empowered Korean Methodist women worked hard to achieve their own rights in their homes and in churches and society, and contributed to the evangelization of the people, the establishment of Christian homes, and the modernization, democratization, and independence of the nation. The gendered side of modernization and nationalism came through the mission of Southern Methodist female missionaries, as seen in the 1919 Independence Movement. Korean Methodist women were able to contextualize the Christian message, found and built up their own identities, and became the subjects of history who transformed themselves, their families, church, communities, nation, and foreign lands.

**Author Contributions:** Writing—original draft, A.S.-V. and M.I. All authors have read and agreed to the published version of the manuscript.

**Funding:** This research received no external funding.

**Institutional Review Board Statement:** Not applicable.

**Informed Consent Statement:** Not applicable.

**Data Availability Statement:** Not applicable.

**Conflicts of Interest:** The authors declare no conflict of interest.

## Note

1. "Miss Mattie M. Ivey in Wonsan," *ARWMCMECS*, 1905–1906, 47; *ARWMCMECS*, 1906–1907, 41–42; *ARWMCMECS*, 1907–1908, 40; Miss Mamie D. Myers, "Lucy Cuninggim School," *ARWMCMECS*, 1910–1911, 308–309; Miss Lillian Nichols, "Carolina Institute," *ARWMCMECS*, 1911–1912, 261–263; Miss Ellasue Wagner, "Holston Institute," *ARWMCMECS*, 1911–1912, 268–269; *ARWMCMECS*, 1913–1914, 206; Mrs. J.P. Campbell, "Seoul—Woman's Work, Water Mark and Water Gate Churches," *ARWMCMECS*, 1914–1915, 203–204; *ARWMCMECS*, 1915–1916, 73–75; Miss Ellasue Wagner and Miss Lillian Nichols, "Holston Institute," *ARWMCMECS*, 1917–1918, 222–224; "Holston Institute, at Songdo," *ARWMCMECS*, 1918–1919, 96–98; Miss Ida Hankins, "Mary Helm Industrial Department of Holston Institute," *ARWMCMECS*, 1918–1919, 373–374; Miss Hallie Buie, "Lucy Cuninggim Industrial School," *ARWMCMECS*, 1918–1919, 376–378; Miss Ida Hankins, "Mary Helm Industrial Department of Holston Institue," *ARWMCMECS*, 1919–1920, 107–110. The social-evangelistic centers also provided classes related to industrial arts. Agnes Elise Graham, "Social-Evangelistic Center, Songdo," *ARWMCMECS*, 1922–1923, 296–302.

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
