# Peer review of "The World Was Their Parish: Evangelistic Work of the Single Female Missionaries from the Methodist Episcopal Church, South, to Korea, 1887–1940"

_religions, doi:10.3390/rel14020262_

Round 1
Reviewer 1 Report
This is generally a very well done article, ready to be published with one reservation.
The positives: well written, well-organized, good use of previous research; provides clear connections between the “Evangelistic Work of the Single Female Missionaries” and the social/political development of Korea from the late 19th century to 1940.
All good.
But.
Perhaps the author(s) work(s) comfortably within a USAmerican culture, familiar with the history of slavery and racism. This reviewer is Canadian, and is left wondering how anything done in the context of accepting slavery can be called “the service of God” (124). From the perspective of a non-American sociologist of religion it is passing strange that a history of female enterprise and empowerment in the context of conflict over slavery and racism can be presented without reference to slavery and racism. Was the MECS not an “ethnic church” – a denomination of white people? This – or a statement of whatever the reality was – should be made clear for an international audience. Did the leaders of the SM women’s mission organizations simply accept the dis-empowerment of African Americans (structural violence, torture, rape, murder, exploitation, intimidation, discrimination, dis-privilege) while seeking their own agency and empowering women in Korea, Koreans generally and Korean society?
What should an outsider-to-American-society know about the MECS and race/slavery in the various decades of its history?
Please clarify the issue of racism.
The sentences after line 150 seem to suggest a basic un-addressed contradiction: the acceptance of patriarchy, racism, even slavery, and rigid social stratification in the context a gospel worldview. Despite Paul and Galatians, it seems to be an attitude that in Christ there IS male and female, slave and free.
This is good research, well-reported. I hope it is published. But for an outsider to American society and history, it seems to have contradictions. Perhaps it is faithfully reporting real social contradictions. Please clarify.
Author Response
Dear Reviewer, I appreciate the comments and concerns for clarity in the issue you raised about slavery. I added this paragraph from 124 to 144. I hope this clarifies your concerns. peace
The female missionaries were women of their times, and it took time for them to denounce slavery. It was not until 1897 that Bennett began mentioning the demand for organized missionary work among blacks, and continually encouraged the Methodist women to get involved with interracial settlement work. She stated, “There are two large classes of society touching our lives every day, whether we be in the country, in the towns, or in the large cities, to which our Society has yet given no helping hand. We have done nothing for the negroes. Our great Church, as a Church, has done but little. Individual members may, doubtless, have done much; but as an organization the path of duty is plain before us…. We must enterprise some special work for this great race of people” (Bennett). In 1912, the WMC members at last responded to Bennett’s call to action by launching settlement work for blacks, and in 1913 founded the Nashville Bethlehem House in Nashville, TN. The Woman’s Board also established additional Bethlehem Houses in Augusta, Georgia, Birmingham, Alabama, and Chattanooga, TN. The Southern Methodist women in the North America soon began to extend their contribution more to the public realm by employing an all-out social gospel program that emanated in a strong social concern for the marginalized of the society. In 1916, the project took account of “the abolition of child labor, the reduction of illiteracy, prison reform, and end to the convict lease system, the cultivation of ‘sympathy between all races,’ and the solution of race problems in a spirit of helpfulness and justice” (Scott 143).
Reviewer 2 Report
The article clearly presents the success of the women missionary of the Methodist order in South Korea. Yet, although a well written description of the history of religions in Korea, the article looks more like a pamphlet of advertisement of the order, rather than a scholarly article. In order that this article will look more like a scholarly article, I would have expected to see comparative information with other orders. I would like to learn more about religious challenges in Korea. What about secularity in South Korea in our secular age? I understand that the women missionary of the Methodist order are great, but what do we learn from their achievements on the Korean society? I can't expect an answer to all the questions, but would like to see at least some essential study and not only a report.
Author Response
Thanks for your comments. I am not sure how to respond to the criticisms. The paper is a historical piece and should not cover "secularity in South Korea in our secular age." Why should I imposed a modern theory to an ancient context? It is not good historiography.
Their achievements to Korean society are quite clear in the paper: they established the first school for girls and boys, the first university for women, the first women's hospital, the first school for the blind, and were influential teachers of the Korean women who were at the 1919 independence showdown with the Japanese.
The paper is about the single missionaries of the Methodist Episcopal Church, South. A comparison to another denomination would be for a book.
Reviewer 3 Report
line 35: WFMM is an unclear acronym, similar to enough of them that spelling it out would be useful.
ll 46-51: I like the way that the thesis of this section clearly articulates the meaningfulness of the backstory in terms of the larger argument.
ll 70-85: This is largely an understandable list of names/dates, due diligence for context, concise enough to not distract from the ongoing argument.
97-106: I appreciate how you weave in the components of the theological/religious landscape that were empowering to individuals.
107-116: ...and then bring this back to the main thesis. Amazing!
120-121: "promised covenant with God" should be reworded
131: Unclear what the "six" departments are
144-156: I again admire how you're able to take 21st century insights into the far-reaching and damaging influences of patriarchy and express their relevance toward your thesis.
157-190: Great section!
192-253: You do a great job of synthesizing enough details about religious influences to allow unfamiliar readers to have a basic orientation to Korea's religious environment while also providing treatments of diverse traditions that educated readers understand as fair. It's a tough balance: nice work.
267, 269, 284, 285, 295, etc. "Gospel" and "she/her" stand out as descriptors that could be perhaps replaced with more neutral terms (Christian doctrine, it/its).
254-295: Once again, you admirably synthesize a lot of history into the most meaningful details, weaving them into a story that your reader can follow mindful of the context it is designed to offer.
317-323 should be a pulled quote
338-346: I again admire your clear work of integration demonstrated in paragraphs like this one.
364-371: Returning to the source of holism is an important reminder here.
460-470: This has been a riveting section, and the paragraph at this point provides the useful function of continuing to orient your reader to the larger thesis of the holistic nature of the approach.
572-580: Make sure this is a pulled quote, indented.
581-592: You set up this thesis re-statement well: your reader is well-positioned to accept it.
626 (and throughout): Paewha is in a different font and should conform to the rest of the text. 645-648 is similarly in need of conforming.
Author Response
Thanks for all the comments.
reworded 120-21
deleted "six departments" 131
changed Gospel to Christian doctrine or Christian message in 267, 269, 284, 285, 295.
changed Pewha to conform to font in 626